

# Transcriptome reveals differential expression of flavor and color in closely related strains of tomato (*Solanum lycopersicum*)

Chunmei Guo[1,*], Xiuyuan Liu[1,*], Yingfeng Ding[1], Zhaoyilan He[1], Songmei Shi[1], Yumei Ding[2], Hui Shen[1] and Zhengan Yang[1]

[1] College of Landscape Architecture and Horticulture, Yunnan Agriculture University, Kunming, Yunnan, China
[2] College of Food Science and Technology, Yunnan Agriculture University, Kunming, Yunnan, China
* These authors contributed equally to this work.

Corresponding authors
Hui Shen, hbshenhui@163.com
Zhengan Yang,
yangzhengan@ynau.edu.cn

## ABSTRACT

Fruit flavor and color are critical quality characteristics of tomatoes (*Solanum lycopersicum*). Numerous studies have demonstrated that tomato flavor is primarily linked to the sugar-acid content and its ratio, while fruit color is predominantly determined by the composition and concentration of carotenoids and flavonoids. To elucidate the regulatory mechanisms underlying the differences in sugar-acid and color formation during fruit ripening, transcriptome analysis was conducted for the first time on the breaker stage (Br) and mature fruit stage (MF) of the closely related strains yellow-fruited tomato (No. 19) and red-fruited tomato (No. 20). This analysis aimed to identify key regulatory genes and biosynthetic pathways associated with the formation of flavor and color in tomato fruits. The transcriptome analysis revealed that 1,546 differentially expressed genes (DEGs) were identified in the Br19_vs_Br20 comparison, of which 507 were up-regulated and 1,039 were down-regulated. In the MF19_vs_MF20 comparison, 2,178 DEGs were detected, with 1,235 up-regulated and 943 down-regulated. Upon further analysis of the DEGs, we identified several key genes in the sugar-acid metabolic pathway, including sucrose synthase (SUS), phosphofructokinase (PFK), fructose-bisphosphate aldolase (FBA), citrate synthase (CS), and succinate dehydrogenase (SuDH), which may significantly influence tomato flavor. Additionally, differential genes related to carotenoid and flavonoid metabolism, such as cytochrome P450 98A (CYP98A), caffeoyl-CoA3-O-methyltransferase (CCoAMT), carotenoid isomerase (CRTISO), lycopene beta cyclase (LCYB), zeaxanthin epoxidase (ZEP), violaxanthin deepoxidase (VDE), and 9-cis-epoxycarotenoid dioxygenase (NCED), as well as genes linked to ethylene synthesis, such as 1-aminocyclopropane-1-carboxylate synthase (ACS) and 1-aminocyclopropane-1-carboxylic acid oxidase (ACO), may play a role in the color changes observed in tomatoes. The findings of this study provide insights into the underlying mechanisms of flavor and color development in tomato fruit, offering valuable information for the genetic improvement of tomatoes.

## INTRODUCTION

Tomato (*Solanum lycopersicum*) is one of the principal vegetable crops cultivated in agricultural facilities in China (*Kurina et al., 2021*). An increasing number of consumers are being drawn to its unique flavor and rich nutritional profile (*Li, Liu & Fu, 2023*). The quality of tomatoes is primarily assessed based on appearance, nutritional value, and flavor quality (*Li et al., 2022*). Specifically, flavor quality encompasses attributes such as sweetness, acidity, and aroma, which are linked to the presence of soluble sugars, organic acids, and volatile aromatic compounds in the fruit. The appearance quality of tomatoes includes factors such as fruit size, shape, color, and firmness. This aspect significantly influences consumer purchasing decisions and serves as a key indicator for evaluating overall tomato quality.

The flavor quality of tomato fruits is a complex trait primarily influencing consumer preference. It mainly includes the contents of soluble solids, sugars, organic acids, and volatile aromatic substances (*Gascuel et al., 2017*). The pleasant taste of tomatoes is determined by an appropriate sugar-to-acid ratio (*Ye et al., 2025*). Glycolysis is one of the important pathways of sugar metabolism in living organisms, and the activity of key enzymes in glycolysis, such as hexokinase (HK), phosphofructokinase (PFK), and pyruvate kinase (PK), directly influences the rate of glycolysis, thereby regulating the supply of precursors for sweet-tasting substances (*Zhang et al., 2024*). In tomato fruits, sugars primarily accumulate in the form of fructose and glucose. During fruit ripening, the products of photosynthesis in leaves are continuously transported to the fruits in the form of sucrose, which is then broken down into fructose and glucose by enzymes such as sucrose synthase (SUS) and acid invertase (AI), leading to the continuous accumulation of sugars in the fruits (*Tieman et al., 2017*). The sourness of tomatoes primarily stems from the organic acids contained in their fruits, particularly citric acid and malic acid. The presence of these organic acids in tomatoes mainly relies on the tricarboxylic acid (TCA) cycle in mitochondria. The TCA cycle influences the sourness of tomatoes by participating in the synthesis of organic acids in tomato fruits (*Xu et al., 2012*). The key enzymes of the TCA cycle in tomatoes include citrate synthase (CS), isocitrate dehydrogenase (IDH), and the pyruvate dehydrogenase complex (PDHe) (*Guillet et al., 2002*). With precise determination of soluble sugars and organic acids in tomato fruits, numerous gene loci associated with these components have been identified. For example, *Tieman et al. (2017)* analyzed the flavor substance contents of over 400 tomato materials, followed by whole-genome sequencing and genome-wide association analysis, identifying five gene loci related to acid content and two related to sugar content.

The color of tomato fruits, a crucial quality trait, is attributed to the accumulation of secondary metabolites, specifically carotenoids and flavonoids (*D'Esposito et al., 2024*). Carotenoids are the primary pigments in tomatoes, and their content and variety significantly influence the fruit's color (*Jiang et al., 2024*). Phytoene synthase (PSY) is a key enzyme in carotenoid metabolism and plays an essential role in the carotenoid

biosynthesis pathway. Fraser demonstrated that overexpressing the *PSY* gene in tomatoes resulted in a 2–4 fold increase in the total carotenoid content of the fruits (*Fraser et al., 2002*). Specifically, the levels of lycopene, β-carotene, and lutein were enhanced by 2.4, 1.8, and 2.2 times, respectively. Phytoene is enzymatically converted into lycopene through the actions of phytoene dehydrogenase (PDS) and ζ-carotene dehydrogenase (ZDS). Lycopene is further transformed into β-carotene and α-carotene through the catalytic actions of lycopene β-cyclase (LCYB) and lycopene ε-cyclase (LCYE) (*Nisar et al., 2015*). Subsequently, β-carotene and α-carotene are converted into zeaxanthin and lutein by zeaxanthin epoxidase (ZEP) and violaxanthin de-epoxidase (VDE), respectively (*Lu & Li, 2008*). Ethylene promotes tomato fruit ripening and carotenoid accumulation. As a typical climacteric fruit, ethylene regulates the formation of color, texture, and flavor compounds during the ripening process of tomato fruits (*Schijlen et al., 2007*). In tomato fruits, the biosynthesis of ethylene is primarily catalyzed by two enzymes: first, SAM (S-adenosylmethionine) is converted into ACC (1-aminocyclopropane-1-carboxylic acid) by the action of ACS (1-aminocyclopropane-1-carboxylic acid synthase), and then ACC is transformed into ethylene through the activity of ACC oxidase (*Fraser, Bramley & Seymour, 2001*). Existing research has confirmed that in the fruit ripening regulatory network, ethylene, as a key signaling molecule, can positively activate the expression of the PSY gene, thereby driving the specific accumulation of lycopene in fruit tissues.

Flavonoids are among the primary pigments found in plants, contributing a diverse array of colors (*Cavaiuolo, Cocetta & Ferrante, 2013*). These compounds are biosynthesized from phenylalanine *via* the phenylpropanoid pathway, where phenylalanine is converted into p-coumaroyl-CoA through the actions of phenylalanine ammonia-lyase (PAL), cinnamate 4-hydroxylase (C4H), and 4-coumarate: CoA ligase (4CL) (*Wang et al., 2018b*). The entry of p-coumaroyl-CoA into the flavonoid biosynthesis pathway marks the initiation of specific flavonoid synthesis, beginning with the formation of chalcone (*Nabavi et al., 2018*). p-Coumaroyl-CoA is enzymatically reduced by chalcone synthase (CHS) to produce naringenin chalcone and chalcone, which are subsequently isomerized to flavone by chalcone flavanone isomerase (CHI) (*Anwar et al., 2019*). Chalcone, recognized as the first critical intermediate in the flavonoid metabolic pathway, provides the essential scaffold for downstream flavonoid synthesis and also acts as an important yellow pigment in plants (*Zou et al., 2022*). *CHS*, a polyketide synthase, is the crucial initial rate-limiting enzyme in the flavonoid biosynthesis pathway (*Deng et al., 2013*; *Zhang et al., 2017*). In tomatoes, the reduction of *CHS* through RNA interference (RNAi) results in a decrease in total flavonoid content (*Schijlen et al., 2007*).

In the present study, we examine two closely related strains, No. 19 and No. 20. The peel of No. 19 is characterized by a yellow color, whereas No. 20 exhibits a red peel. Additionally, No. 19 has a higher sweetness level and lower acidity compared to No. 20. Aside from these differences in fruit color and sugar-acid content, other traits between No. 19 and No. 20 show no significant variation. After years of observation and evaluation, these traits have proven to be stable. However, the molecular mechanisms underlying the differences in traits between these two tomato strains remain inadequately explored. Therefore, No. 19 and No. 20 serve as ideal materials for investigating the mechanisms

behind synthesis disparities in tomatoes with varying qualities. In this research, we initially assessed the physiological indices of the fruits from both strains, revealing significant differences in the contents of lycopene, soluble solids, soluble sugars, and total acids. Subsequently, transcriptome analyses were performed during the breaker and maturity stages of the fruits of No. 19 and No. 20. Through enrichment analysis and metabolite-gene correlation analysis, we identified genes associated with the synthesis of sugars, acids, and colors during the development of tomatoes with differing qualities. The identified genes were further validated using quantitative polymerase chain reaction (qPCR). This study provides essential insights into the formation and variation of flavor and color in tomato fruits, which will facilitate the development of tomato varieties with specific flavor and appearance quality characteristics.

# MATERIALS AND METHODS

## Experimental materials and sampling

Tomato seeds of strains No. 19 and No. 20 were provided by the Vegetable Research Group of the College of Landscape and Horticulture at Yunnan Agricultural University. The plants were sown on April 1, 2023, in the vegetable glass greenhouse at Yunnan Agricultural University. Seedlings of the two strains were transplanted from the seedling tray to the trough cultivation field on April 30, under conditions of a daily temperature averaging 25 ± 3 °C and a relative humidity of 50 ± 10%. Fertilization was administered every three days, utilizing the Hoagland universal water formula as the nutrient solution. Notable differences in fruit flavor and color were observed between tomato No. 19 and tomato No. 20, while no significant differences were detected in other traits, which have remained consistent over years of observation and evaluation. Fruits exhibiting consistent growth, free from mechanical damage, pests, and diseases were selected as test materials at two distinct periods: the color-breaking stage (Br, 40 days after flowering) and the maturity stage (MF, 55 days after flowering). Samples were taken from the pericarp and pulp of the circum-equatorial region of the fruits, with three biological replicates for each period, and five fruits per replicate. These samples were quickly frozen in liquid nitrogen and subsequently stored at −80 °C for future analysis.

## Measurement of tomato quality indexes

Total acid content was determined by acid-base neutralization titration (*Fraser, Bramley & Seymour, 2001*). A total of 5.0 g of fruit homogenate was weighed 50 mL of ultrapure water was added, and ultrasonic extraction (40 kHz, 25 °C, 15 min) was performed, followed by centrifugation (8,000 rpm, 10 min) and filtration. This was titrated with 0.1 mol/L NaOH standard solution, using 1% phenolphthalein as an indicator, and the results were expressed as citric acid equivalent (g/100 g FW). Vitamin C (VC) was determined according to the 2,6-dichlorophenolindophenol method (*Fraser, Bramley & Seymour, 2001*). A total of 20 mL of 2% oxalic acid solution was added to 2.0 g of fruit pulp, ground in an ice bath, diluted to 50 mL, and centrifuged (4 °C, 10,000 rpm, 15 min). The supernatant was reacted with 2,6-dichloroindophenol sodium salt (≥95%; Sigma-Aldrich, Burlington, MA, USA) for color development, and the absorbance was measured at 520 nm. The

content (mg/100 g FW) was calculated using the standard curve method. Soluble sugars were determined by the anthrone colorimetric method (*Fraser, Bramley & Seymour, 2001*). A 3.0 g sample was refluxed with 80% ethanol (10 mL) in a water bath at 80 °C for 30 min, followed by centrifugation and repeated extraction twice. The supernatants were combined and diluted to 25 mL. A 1.0 mL aliquot of the extract was reacted with 0.2% anthrone-sulfuric acid reagent in a boiling water bath for 10 min for color development, and the absorbance was measured at 620 nm. The content (g/100 g DW) was calculated using a glucose standard curve. Lycopene: Extracted by the ethyl acetate method (*Arya, Mahajan & Jain, 2000*). A 1.0 g of pulp was weighed 10 mL of ethyl acetate-acetone (2:1, containing 0.05% BHT) was added, and shaken in the dark for 30 min. After centrifugation (8,000 rpm, 10 min), the extraction was repeated until colorless. The extracts and measure were combined at 503 nm using a UV-1800 spectrophotometer (Shimadzu, Kyoto, Japan). The content (mg/kg FW) was calculated based on the extinction coefficient ($\varepsilon = 172$). Soluble solids content (SSC) were measured using a PAL-1 refractometer (ATAGO, Tokyo, Japan). The juice was placed on the prism surface, calibrated at 25 °C, and Brix value (%) was read (*Welsch et al., 2007*). Each treatment's quality indicators were conducted with three replicates, with five fruits per replicate.

## RNA-seq analysis

Transcriptome sequencing was commissioned from Lianchuan Biotechnology Co. (Hangzhou, China). Total RNA was extracted from frozen tomato tissue using standard extraction methods. The RNA-seq experimental design consisted of three biological replicates, and each sample was processed and analyzed independently without technical duplicates. The RNA samples underwent rigorous quality control using the Agilent 2100 Bioanalyzer. Libraries were constructed with the NEBNext Ultra Directional RNA Library Prep Kit for Illumina, and the mRNA libraries from each sample were subsequently sequenced using Illumina technology. Low-quality reads, N-containing reads, and spliced reads were removed from the sequenced data. The Q20, Q30, and GC contents of the clean data were then calculated. HISAT2 was employed to map the paired-end clean reads to the reference genome (SL3.0 GCA_000188115.3) for comparison. Based on the comparison result file, the expression levels of all transcripts were estimated using StringTie (*Pertea et al., 2016*) by calculating fragments per kilobase of exon per million fragments mapped (FPKM). Differentially expressed genes (DEGs) were analyzed using the DESeq2 package in R, with a screening threshold of $|\log2(FoldChange)| > 1.0$ and a q value $< 0.05$. Gene Ontology (GO) and Kyoto Encyclopedia of Genes and Genomes (KEGG) enrichment analyses were performed using the clusterProfiler package. The statistical power of this experimental design, calculated in RNASeqPower is 0.8.

## qPCR analysis

We selected 31 genes associated with sugar acid metabolism and fruit color for qPCR validation. cDNA was synthesized from total RNA using the PrimeScript™ RT Reagent Kit with gDNA Eraser (Perfect Real Time) (TaKaRa, Dalian, China), following the manufacturer's instructions. Actin served as the internal reference gene. The qPCR

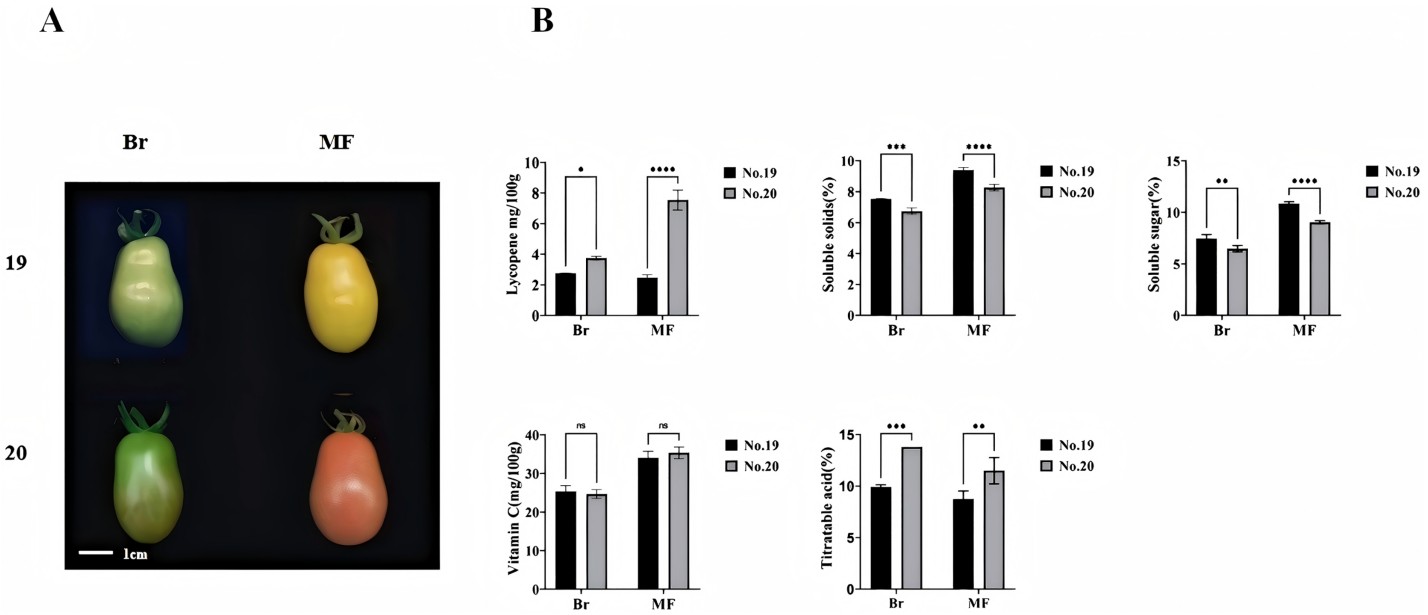

**Figure 1 Phenotypes and lycopene content of tomato fruits from strains 19 and 20 at the color breaking and maturity stages.** (A) Phenotype (B) soluble solids, soluble sugars, titratable acids, vitamin C, and lycopene content. Br, breaking stage; MF, maturity stage. All data are expressed as means of three biological replicates, with error bars indicating standard deviation. Asterisks denote significant differences between the two tomato strains at the same stage: *$p < 0.05$, **$p < 0.01$, ***$p < 0.001$, ****$p < 0.0001$.

analysis for each sample was conducted in triplicate. Primers were designed using Primer Premier 5.0 software (Table S1). All qPCR data were analyzed with GraphPad Prism 9.0 (GraphPad Software, San Diego, CA, USA).

## RESULTS

### Analysis of physiological indicators in tomato fruits

We measured the physiological indicators of No. 19 and No. 20 at both the breaking stage and the red maturity stage (Fig. 1A). Significant differences were observed in the contents of lycopene, soluble sugar, soluble solids, and total acids between the two tomato strains at these stages. Notably, the lycopene content was lowest in No. 19 during the mature period and highest in No. 20, with an evident increase of 67.11% in No. 20 compared to No. 19. The soluble sugar content in No. 19 increased significantly by 11.89% and 13.44%, respectively, at the two stages when compared to No. 20. Similarly, the soluble solids content in No. 19 showed significant increases of 15.46% and 8.54% at the two stages relative to No. 20. Furthermore, the total acid content in No. 19 was substantially reduced by 30.43% and 31.58% at the breaking color and maturity stages, respectively, compared to No. 20. However, there was no significant difference in vitamin C content between No. 19 and No. 20 at either stage (Fig. 1B).

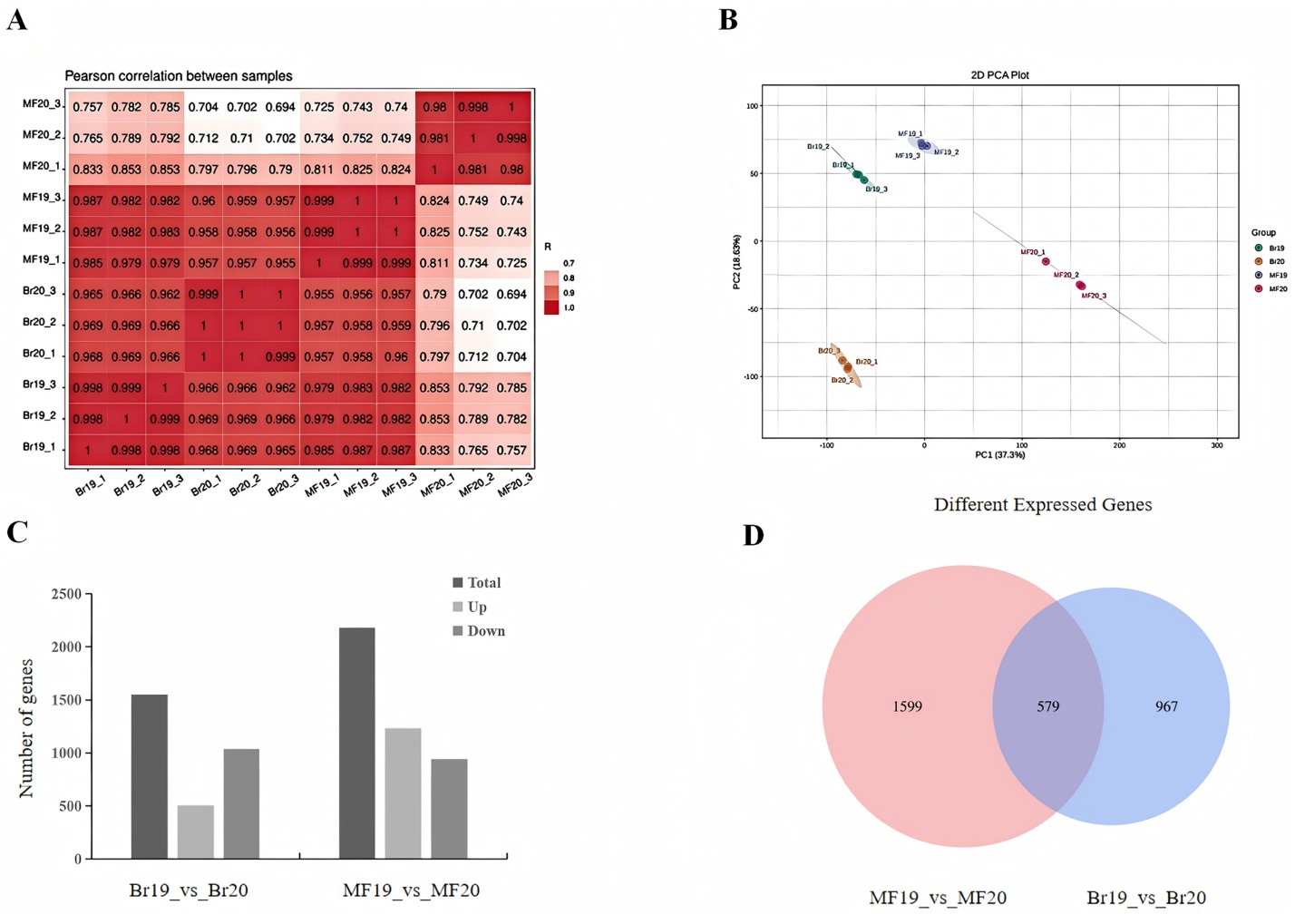

**Figure 2  Number of DEGs in two tomato strains.** (A) Sample correlation plot. (B) PCA score plot. (C) DEGs in different comparison groups. (D) Venn diagram of DEGs in different comparison groups.     

## Transcriptome analysis

### Identification of differentially expressed genes

To further investigate the factors contributing to the differences in sugar-acid and color of tomatoes at two developmental stages, transcriptome sequencing was conducted on the breaking stage and the maturity stage of No. 19 and No. 20 strains. A total of 12 cDNA libraries were constructed, with three replicates for each of the two developmental stages. Following the filtering of raw reads to eliminate low-quality data, the valid bases for each sample reached 5 Gb, with a Q20 value exceeding 98%, a Q30 value exceeding 92%, and a GC content ranging from 42% to 42.5%. The sequencing results were aligned with the reference genome, achieving an alignment efficiency greater than 90%. These data indicators collectively suggest that the quality of the sequencing data is sufficient for subsequent analysis (Table S2). The Pearson correlation coefficients among samples were all greater than 0.998, indicating strong repeatability and a positive correlation in gene expression (Fig. 2A). We employed principal component analysis (PCA) to assess the

overall differences in gene expression between the fruits at different stages of the two strains. The PCA results demonstrated clear separation of the three replicates for each sample type, indicating high consistency and quality of the data. PC1 and PC2 accounted for 55.93% of the total variation, with the first principal component (PC1) alone explaining 37.3% of the variance. This finding indicates significant differences between the two tomato strains at the two developmental stages (Fig. 2B). By comparing the differences between samples at different developmental stages, DEGs were screened using the methods mentioned in the Materials and Methods section. When comparing the samples of No. 19 and No. 20 at the same developmental stage, it was observed that at the breaker stage, 1,546 DEGs were identified, with 507 being up-regulated and 1,039 being down-regulated. This indicates that the majority of DEGs exhibited a down-regulated pattern in tomato during the breaker stage (Fig. 2C, Table S3). At the mature stage, 2,178 DEGs were detected, among which 1,235 were up-regulated and 943 were down-regulated (Fig. 2C, Table S4). In total, 579 genes were differentially expressed in both comparison groups. There were 967 genes uniquely differentially expressed at the breaker stage and 1,599 genes uniquely differentially expressed at the mature fruit stage (Fig. 2D).

### Functional enrichment analysis of differentially expressed genes

To clarify the primary biological functions and regulatory mechanisms of the identified DEGs, GO and KEGG enrichment analyses were conducted. The GO annotations of DEGs detected at the two stages, No. 19 and No. 20, were categorized into three groups: molecular function, biological process, and cellular component. The top 20 GO enrichment results for the comparison between Br19 and Br20 indicated that DEGs were significantly enriched in categories such as iron ion binding (GO:0005506), enzyme activity acting on glycosyl bonds in hydrolases (GO:0016798), and extracellular region (GO:0005576), among others. Similarly, the top 20 GO enrichment results for the comparison between MF19 and MF20 revealed that DEGs were enriched in categories including plastid (GO:0009536), chloroplast thylakoid membrane (GO:0009535), chloroplast (GO:0009507), and thylakoid (GO:0009579), among others (Figs. 3A and 3B and, Tables S5 and S6).

To further investigate the functions of DEGs potentially associated with the formation of sugar-acid content and color in tomatoes, we conducted a functional annotation of DEGs at the breaker and ripe stages for No. 19 and No. 20 using the KEGG database. The analysis of the top 20 entries in the KEGG enrichment results for Br19_vs_Br20 revealed that DEGs were primarily enriched in metabolic pathways, flavonoid biosynthesis, and photosynthesis pathways, along with additional enrichment in carotenoid biosynthesis and the metabolism of amino sugars and nucleotide sugars. Similarly, the analysis of the top 20 entries in the KEGG enrichment results for MF19_vs_MF20 indicated that DEGs were predominantly enriched in photosynthesis, metabolic pathways, and secondary metabolite biosynthesis pathways, with further enrichment observed in flavonoid biosynthesis, carbon metabolism, and the pentose phosphate pathway (Figs. 4C and 4D, Tables S7 and S8).
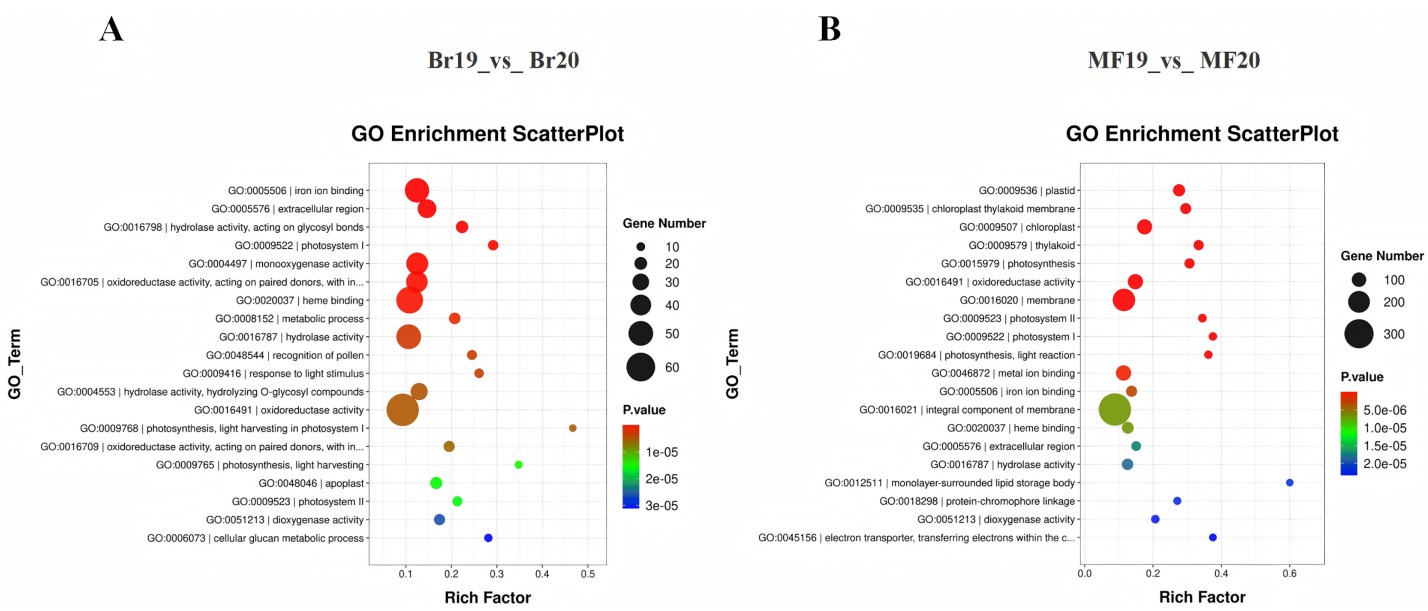

**Figure 3 GO annotation and KEGG enrichment analysis of DEGs.** (A) GO annotation of DEGs for Br19_vs_Br20. (B) GO annotation of DEGs of MF19_vs_MF20.

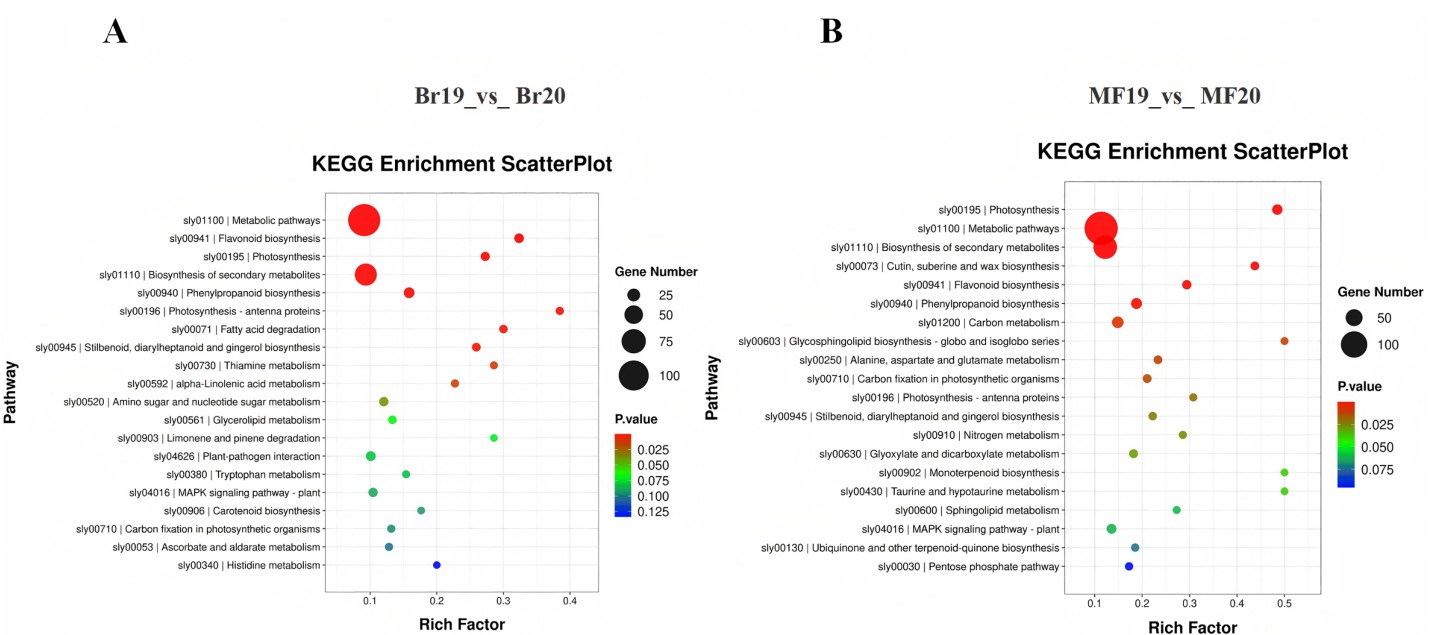

**Figure 4 KEGG enrichment analysis of DEGs.** (A) KEGG enrichment of DEGs for Br19_vs_Br20. (B) KEGG enrichment of DEGs of MF19_vs_MF20.

### *Candidate genes related to sugar-acid metabolism in two tomato strains at different developmental stages*

The correlation analysis between physiological indicators with significant differences and candidate genes suggests that the expression levels of several genes associated with soluble

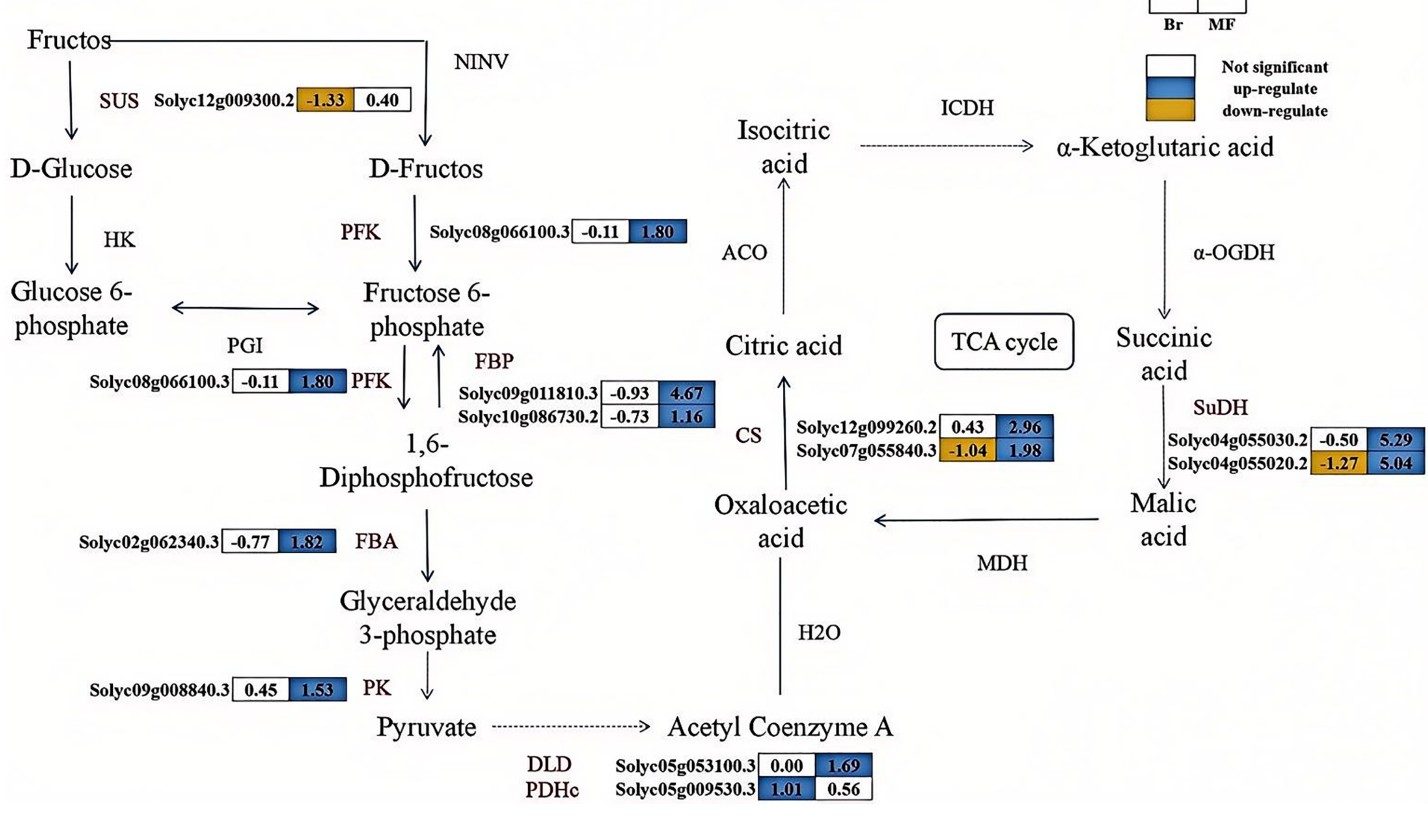

**Figure 5** **The synthesis pathways of sugar-acid metabolism in the fruits of two tomato strains (Br and MF).** The figure depicts the sequential enzymatic reactions in glycolysis and related metabolic processes, with key enzymes highlighted. Key enzymes involved in the pathways are: SUS (sucrose synthase), PFK (phosphofructokinase), FBP (fructose-1,6-bisphosphatase), FBA (fructokinase), PK (pyruvate kinase), DLD (dihydrolipoic acid dehydrogenase), PDHc (pyruvate dehydrogenase complex), CS (citrate synthase), and SuDH (succinate dehydrogenase). Visual annotations: The colored blocks adjacent to gene identifiers (*e.g.*, Solyc12g009300.2) represent gene expression patterns in Br (left block) and MF (right block) strains. The legend at the top right indicates that white blocks signify "Not significant" expression, blue blocks denote "up-regulate", and yellow blocks denote "down-regulate". Solid arrows show direct metabolic conversions catalyzed by the corresponding enzymes (labeled in black, *e.g.*, SUS, PFK). The data in the grid indicate the fold change of gene upregulation or downregulation in one tomato strain compared to the other control strain, expressed as $\log_2$(FoldChange) values. The pathway illustrates the flow of metabolites and the differential gene expression profiles specific to the two tomato strains during sugar-acid metabolism.

sugar and total acid metabolism vary significantly between tomato strains No. 19 and No. 20 (Fig. 5). SUS catalyzes the conversion of sucrose into fructose and glucose, and one differentially expressed *SUS* gene (*Solyc12g009300.2*) was identified. In contrast, the expression level of the *SUS* gene in No. 19 shows an increasing trend as the fruit matures. Fructose-6-phosphate can be converted into fructose-1,6-bisphosphate by PFK. Similarly, fructose-1,6-bisphosphate can be transformed into fructose-6-phosphate by FBP. Subsequently, FBA converts fructose-1,6-bisphosphate into glyceraldehyde-3-phosphate, which is further metabolized by enzymes such as PK to produce pyruvate. A differentially expressed *PFK* (*Solyc08g066100.3*), two differentially expressed *FBPs* (*Solyc09g011810.3*, *Solyc10g086730.2*), one differentially expressed *FBA* (*Solyc02g062340.30*), and one differentially expressed *PK* (*Solyc09g008840.3*) were identified, with their expression levels progressively increasing during the developmental stages of No. 19 fruits. Pyruvate is

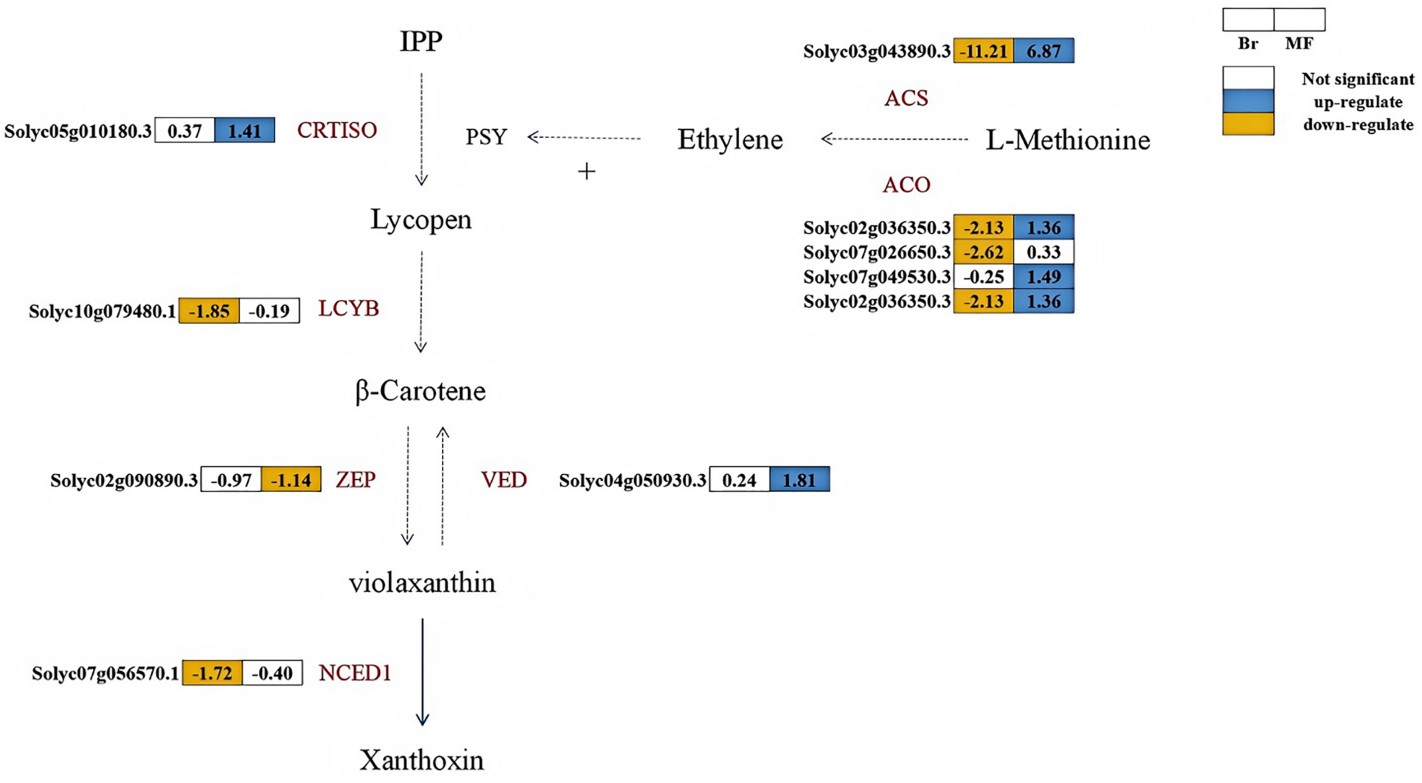

**Figure 6 Metabolic pathways involved in carotenoid and ethylene synthesis in the fruits of two tomato strains (Br and MF).** The figure depicts the core metabolic fluxes, with key enzymes labeled to indicate critical regulatory steps. Key enzymes and their roles are as follows: LCYB (β-cyclase), CRTISO (carotenoid isomerase), VDE (violet xanthophyll de-epoxidase), ZEP (zeaxanthin epoxidase), NCED1 (9-cis-epoxide carotenoid dioxygenase), ACS (1-aminocyclopropane-1-carboxylic acid synthase), ACO (1-aminocyclopropane-1-carboxylic acid oxidase), and SuDH (succinate dehydrogenase). Visual annotations: The colored blocks adjacent to gene identifiers (*e.g.*, Solyc05g010180.3) represent gene expression patterns in Br (left block) and MF (right block) strains. The legend at the top right indicates that white blocks signify "Not significant" expression, blue blocks denote "up-regulate", and yellow blocks denote "down-regulate". Solid arrows show direct metabolic conversions, while dashed arrows represent multi-step reactions or regulatory interactions (*e.g.*, the positive regulation of PSY by ethylene). The pathway structure integrates metabolic relationships and gene expression data specific to the two tomato strains.

subsequently catalyzed by dihydrolipoamide dehydrogenase (DLD) and the pyruvate dehydrogenase complex (PDHe) to form oxaloacetate. The expression of *DLD* (*Solyc05g053100.3*) in sample No. 19 at the mature stage was significantly increased by 1.69 times compared to that in sample No. 20. Additionally, the expression of *PDHe* (*Solyc05g009530.3*) in sample No. 19 at the breaker stage was significantly elevated by 1.01 times relative to sample No. 20. In the tricarboxylic acid (TCA) cycle, oxaloacetic acid is directly converted into citrate through the catalysis of CS. Subsequently, citrate is converted into isocitrate by aconitase (ACO), and isocitrate is further catalyzed by enzymes such as isocitrate dehydrogenase (ICDH) to produce succinic acid, which is then transformed into malic acid through the action of enzymes like SuDH. The expression levels of the *CS* genes (*Solyc07g055840.3, Solyc12g099260.2*) and *SuDH* genes (*Solyc04g055030.2, Solyc04g055020.2*) significantly increase during the fruit development process, indicating that citrate metabolism in tomatoes is enhanced and regulated by these

genes. These findings suggest that these genes play a crucial regulatory role in the balance of sugar and acid components in tomatoes during the ripening process.

### Candidate genes related to carotenoid metabolism and ethylene synthesis in two tomato strains at different developmental stages

A total of six structural genes associated with carotenoid biosynthesis were identified in the fruits of two developmental stages of tomato strains No. 19 and No. 20 (Fig. 6). Lycopene β-cyclase is a key enzyme in the carotenoid synthesis pathway, facilitating the conversion of lycopene into β-carotene. At the breaker stage, the expression levels of *LCYB* (*Solyc10g079480.1*) and *NCED* (*Solyc07g056570.1*) in tomato No. 19 were significantly lower than those in tomato No. 20, with down-regulation observed at 1.85 times and 1.72 times, respectively. CRTISO plays a crucial role in catalyzing the conversion of lycopene from its cis-structure to trans-structure within the carotenoid biosynthesis pathway, while VDE and ZEP are pivotal enzymes regulating the xanthophyll cycle. At the maturity stage, the expression levels of VDE (Solyc04g050930.3) and *CRTISO* (*Solyc05g010180.3*) in No. 19 were significantly higher than those in No. 20, showing up-regulation by 1.81 times and 1.41 times, respectively. Conversely, the expression level of *ZEP* (*Solyc04g051190.3*) in No. 19 was down-regulated by 1.14 times compared to No. 20. The key enzyme phytoene synthase (PSY) in the carotenoid pathway is strongly induced by ethylene during the ripening process, promoting the accumulation of lycopene in fruits (*Alba et al., 2005*). At the breaker stage of No. 19, the expression levels of key enzymes in the ethylene biosynthesis pathway, including *ACS* (*Solyc03g043890.3*), *ACO* (*Solyc07g026650.3*, *Solyc02g036350.3*), were significantly lower than those in tomato No. 20, exhibiting reductions of 11.21, 2.62, and 2.13 times, respectively.

### Candidate genes related to flavonoid metabolism in two tomato strains at different stages

The KEGG enrichment analysis of the transcriptomes from the yellow tomato strain 19 and the red tomato strain 20 revealed that, in both the Br19_vs_Br20 and MF19_vs_MF20 comparisons, the DEGs were significantly enriched in the flavonoid biosynthesis pathway. To evaluate the impact of relevant genes during the color change process in tomatoes, we analyzed the expression of genes associated with flavonoid biosynthesis. In the Br19_vs_Br20 comparison, six genes linked to flavonoid biosynthesis were down-regulated (*Solyc04g063210.3*, *Solyc03g032220.3*, *Solyc05g053550.3*, *Solyc09g091510.3*, *Solyc02g083860.3*, and *Solyc11g013110.2*), while one gene was up-regulated (*Solyc01g009370.2*). At the breaker stage, compared to strain No. 20, the genes related to flavonoid biosynthesis in strain No. 19 exhibited a downward trend. Notably, two *chalcone synthase* (*CHS*) genes, which catalyze the synthesis of naringenin, were significantly down-regulated by 6.95 and 7.13-fold, respectively, in the breaker stage of tomato strain No. 19 compared to strain No. 20. In the MF19_vs_MF20 comparison, three genes were up-regulated (*Solyc03g117600.3*, *Solyc01g009370.2*, *Solyc10g078220.2*), while five genes (*Solyc10g078220.2*, *Solyc05g053550.3*, *Solyc09g091510.3*, *Solyc02g083860.3*, *Solyc11g013110.2*) were down-regulated. At the mature stage, the genes related to

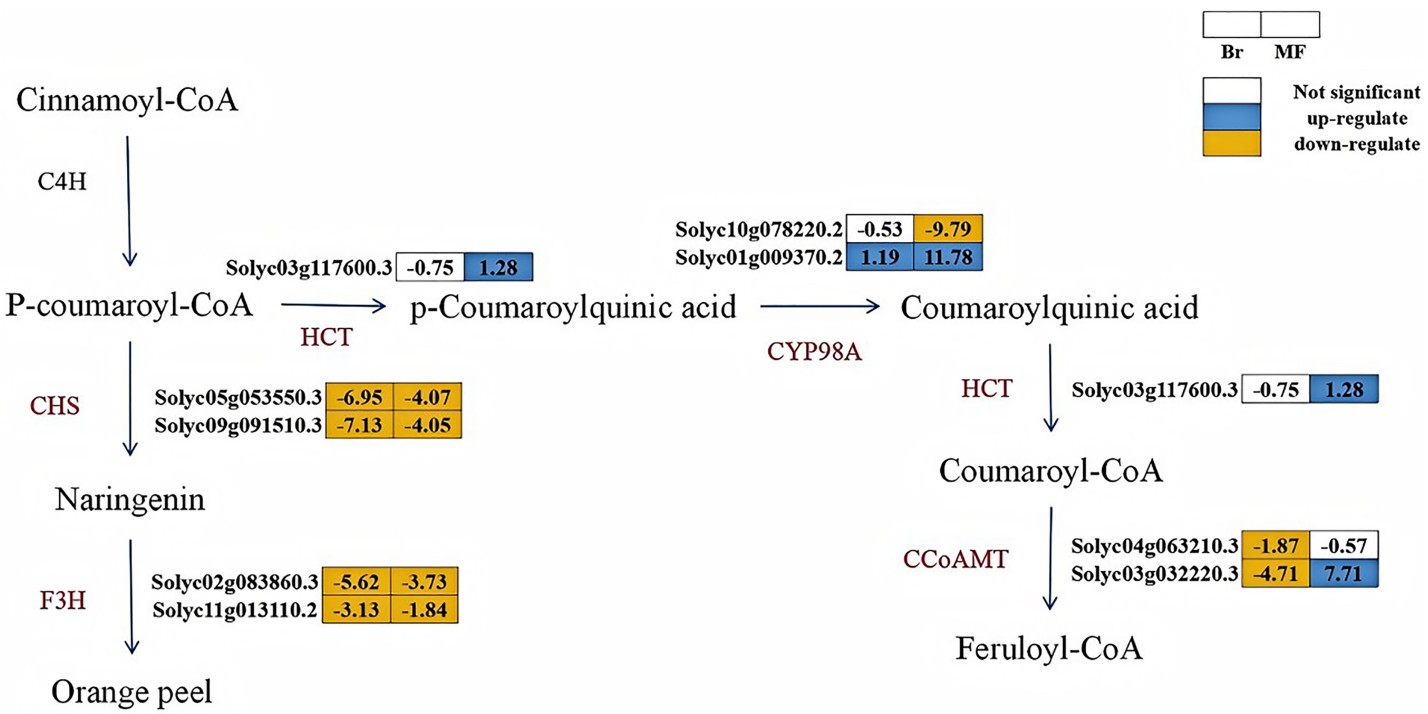

**Figure 7 Metabolic pathways of flavonoid synthesis in the fruits of two tomato strains (Br and MF).** The figure illustrates the sequential enzymatic reactions from Cinnamoyl-CoA to end products like Orange peel, with key enzymes highlighted. Key enzymes involved in the pathways are: CHS (chalcone synthase), F3H (flavanone 3-hydroxylase), HCT (hydroxycinnamoyltransferase), CYP98A (phenol ester 3'-hydroxylase), and CCoAMT (caffeoyl-coenzyme A oxymethyltransferase). Visual annotations: The colored blocks adjacent to gene identifiers (*e.g.*, Solyc03g117600.3) represent gene expression patterns in Br (left block) and MF (right block) strains. The legend at the top right indicates that white blocks signify "Not significant" expression, blue blocks denote "up-regulate", and yellow blocks denote "down-regulate". Solid arrows show direct metabolic conversions catalyzed by the corresponding enzymes (labeled in brown, *e.g.*, HCT, CYP98A). The pathway depicts the flow of metabolites and the differential gene expression profiles specific to the two tomato strains during flavonoid biosynthesis.

flavonoid biosynthesis were significantly up-regulated in strain No. 19 compared to strain No. 20. Among these, the *CYP98A (Solyc01g009370.2)* in strain No. 19 was significantly up-regulated by 11.78-fold at the mature stage compared to strain No. 20, it might lead to the accumulation of coumaroylquinic acid. Additionally, the *CCoAMT (Solyc03g032220.3)* in strain No. 19 also increased significantly by 7.71-fold at the mature stage compared to strain No. 20, which may contribute to the accumulation of feruloyl-CoA content (Fig. 7). Based on the analysis of the aforementioned transcriptome data, we hold that carotenoid metabolism and flavonoid metabolism are of vital importance in the process of tomato color change. In combination with the transcriptomic analysis of the KEGG enrichment pathway, we have elaborated more precisely on the expression of genes associated with flavonoid metabolism and carotenoid metabolism during the development of tomatoes.

### Correlation analysis of fruit soluble sugar, total acid, lycopene contents, and differentially expressed genes

We employed Pearson's correlation coefficient to assess the relationships between DEGs involved in sugar-acid metabolism and carotenoid metabolism pathways, alongside soluble

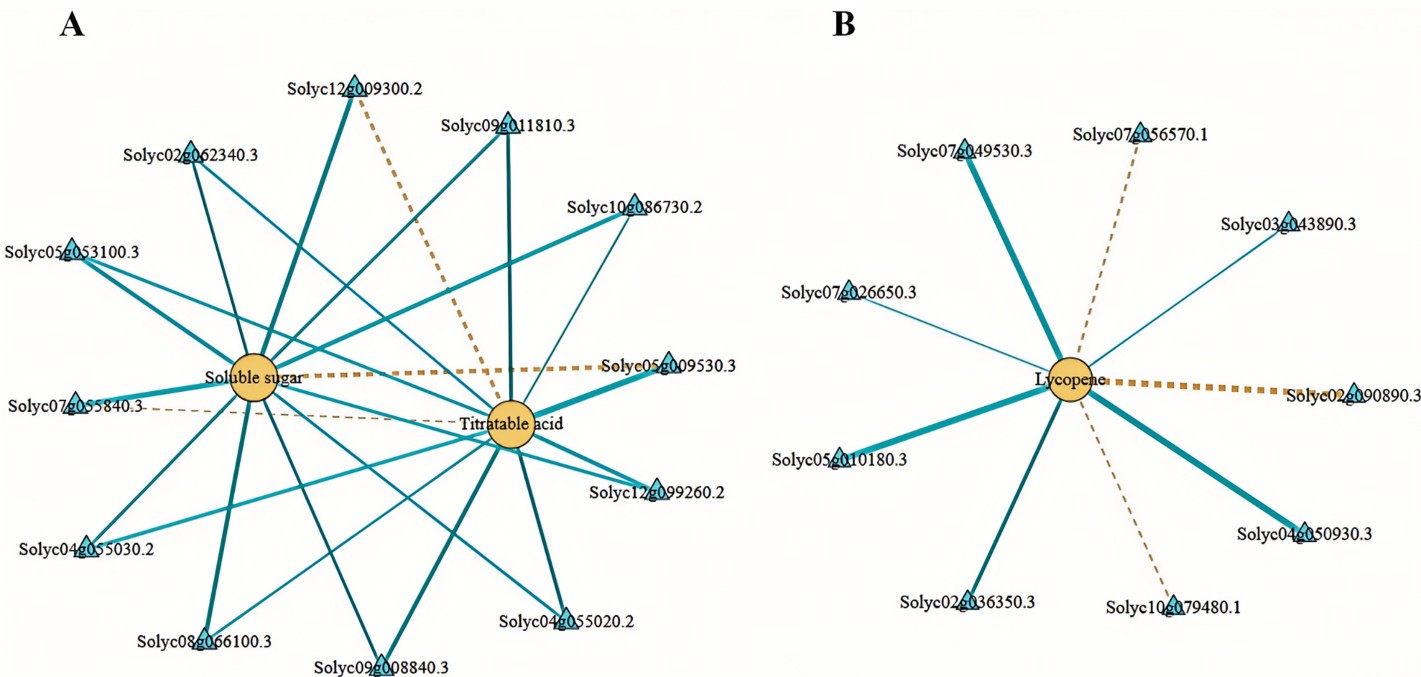

**Figure 8 Network diagram of substances and differential gene correlations involved in sugar-acid metabolism and carotenoid metabolism.** (A) Network diagram of soluble sugars and total acids with associated differential gene correlations. (B) Network diagram of the correlation between lycopene and related differential genes. Yellow circles indicate substances and blue triangles indicate genes. The "yellow dashed strain" and "blue solid strain" indicate positive and negative correlations, respectively, as determined by Pearson correlation coefficients >0.8 or <−0.8 (q-value < 0.1).

sugars, total acids, and lycopene (Fig. 8). In our analysis of sugar metabolites and differential genes, we identified a total of eight DEGs associated with soluble sugars. Among these, six exhibited negative correlations while two displayed positive correlations. Additionally, we identified four DEGs significantly related to total acids in the acid metabolism and synthesis pathways, with three showing negative correlations and one indicating a positive correlation (Fig. 8A). In the correlation analysis pertaining to lycopene, we found nine differential genes linked to carotenoid metabolism, with six negatively correlated and three positively correlated (Fig. 8B). The soluble sugars, total acids, lycopene, and the DEGs discussed above may significantly contribute to the variations in flavor and color observed between tomato varieties No. 19 and No. 20.

### Correlation analysis between differentially expressed structural genes and transcription factors related to tomato fruit quality

In this study, a total of 200 transcription factors were annotated. Upon classifying these transcription factors into families, 35 distinct families were identified. In the breaker stage, the top-ranked transcription factor families by quantity are the ERF family (12 members) and the bHLH family (two members). At the mature stage, the leading transcription factor families remain the ERF family (seven members) and the bHLH family (two members) (Fig. 9A). A correlation analysis was conducted between the identified key structural genes associated with tomato sugar-acid and color synthesis and the transcription factors. The

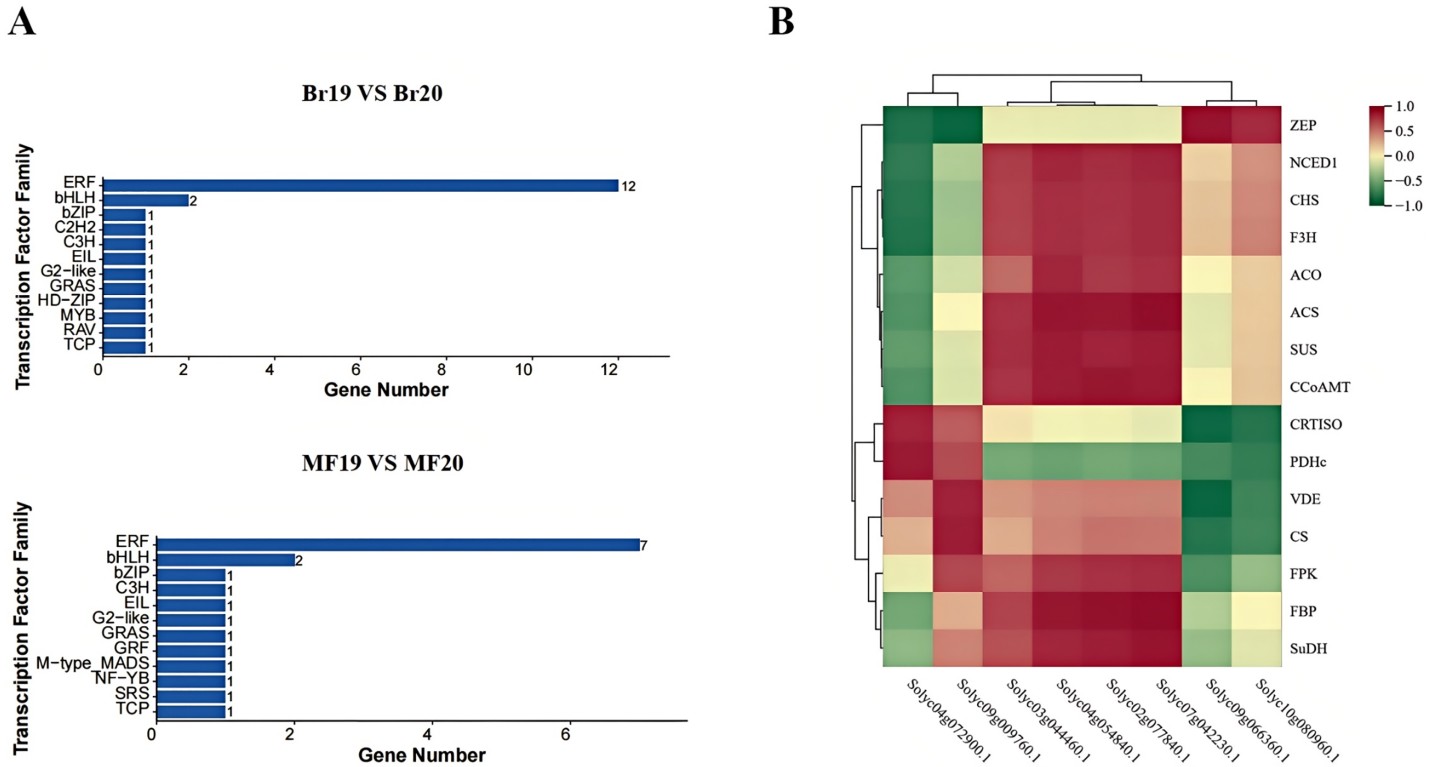

**Figure 9 Transcription factor families and their correlation with key genes at two developmental stages of tomato No. 19 and No. 20.**
(A) Transcription factor families in different control groups; (B) transcription factor families and their correlation with key genes.

results revealed that the genes *CRTISO* and *PDHc* exhibited a significant positive correlation with the transcription factor Solyc04g072900.1. Additionally, transcription factor Solyc09g009760.1 showed a significant positive correlation with the genes *VDE* and *CS*, while demonstrating a significant negative correlation with gene *ZEP*. The genes *ACS*, *CCoAMT*, and *FBP* were significantly positively correlated with the transcription factors Solyc02g077840.1, Solyc04g054840.1, and Solyc07g042230.1. Furthermore, transcription factor Solyc09g066360.1 was significantly positively correlated with gene *ZEP* and significantly negatively correlated with the genes *CRTISO* and *VDE* (Fig. 9B). Expression profile analysis of the screened transcription factors indicated that different transcription factors within the same family displayed both up-regulation and down-regulation at various developmental stages, highlighting the complexity of the fruit ripening process (Table S9).

### qPCR verification of gene expression profiles

To confirm the precision of the RNA-seq data, we carried out a qPCR experiment on the DEGs associated with tomato sugar-acid and color synthesis that were identified (Fig. 10). By comparing the qPCR results with the transcript abundances (FPKM) obtained from transcriptome sequencing, we found that the expression patterns of these genes were
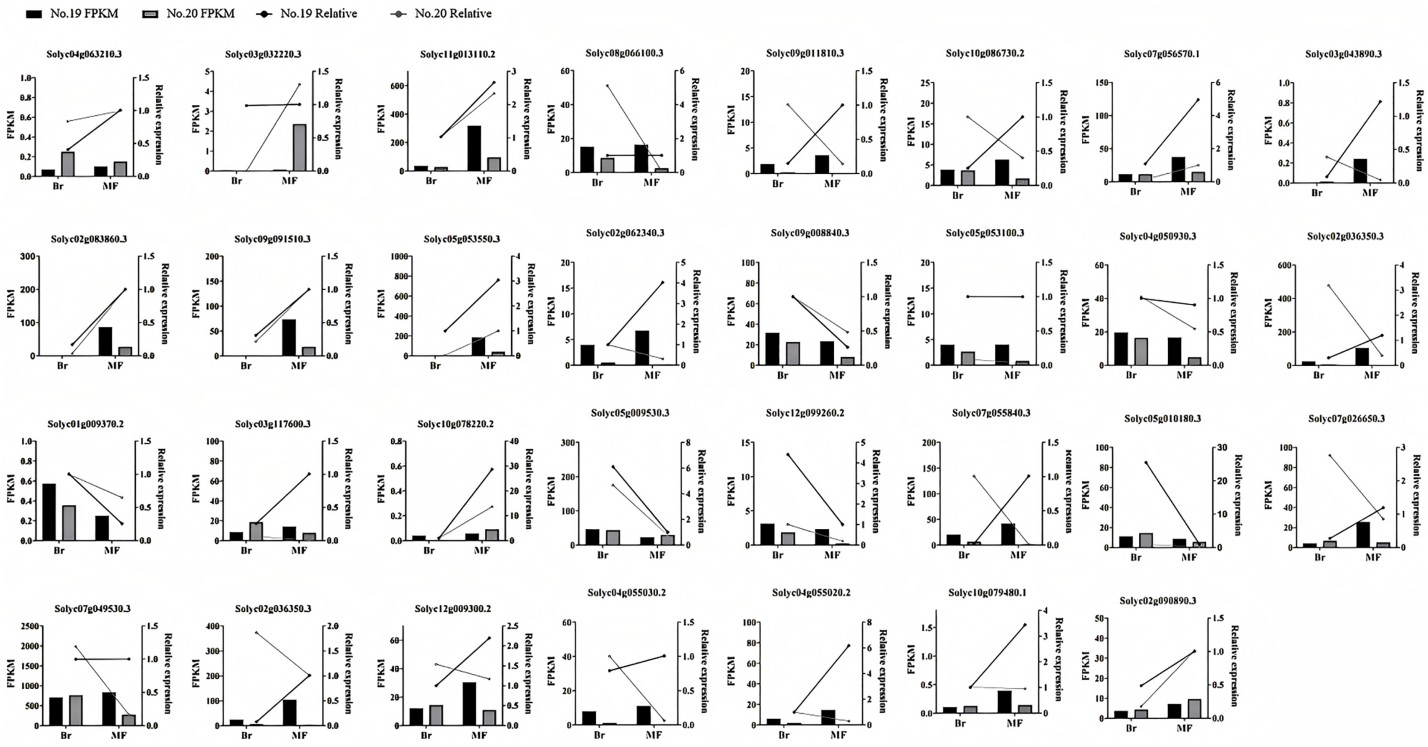

**Figure 10 Expression profiles of significantly DEGs were determined by RNA-seq and qRT PCR.** The histogram represents RNA-seq data (left Y-axis) and the broken line represents qPCR data (right Y-axis).

consistent with the RNA-seq results, indicating that the RNA data was both valid and reliable.

## DISCUSSIONS

The quality of tomatoes is influenced by the interaction of various enzyme systems within the fruit. Tomato fruit compounds are metabolized from sucrose, which enters the fruit (*Yamaki, 2010*). In the presence of sucrose-converting enzymes, sucrose is decomposed into glucose and fructose, which can then be phosphorylated and enter the glycolysis pathway. The glycolytic product, pyruvate, can subsequently enter the tricarboxylic acid cycle, where it participates in the metabolism of citric and malic acids. Pyruvate is further oxidized and decomposed into $CO_2$ and $H_2O$, generating a significant amount of energy that supports various life activities within the fruit (*Xu et al., 2012*). The glycolytic pathway intersects with the mangiferic acid and mevalonate pathways, promoting the synthesis of carotenoids, phenylalanine, linoleic acid, linolenic acid, and branched-chain amino acids (*Wang, Baldwin & Bai, 2016*). As people's living standards improve, there is increasing attention on the quality of tomato fruit, particularly regarding flavor and appearance (*D'Esposito et al., 2017*). Finding ways to enhance the flavor and appearance of tomatoes, while minimizing any adverse effects on their commercial quality, has become a critical focus in tomato cultivation and quality breeding. Furthermore, conducting comparative

analyses between contrasting varieties is an effective approach for identifying differential pathways, expressed genes, or metabolite variants that control important traits (*Fei et al., 2021*; *Wang et al., 2019*; *Xiao et al., 2021*). Therefore, in this study, we employed transcriptomic sequencing to investigate the relevant genes and metabolic pathways that govern changes in sugar-acid content and color in two different strains of tomatoes over time, aiming to preliminarily explore the reasons behind the alterations in the flavor and appearance quality of tomato fruits.

The intensity of tomato flavor is primarily determined by the sugar-acid content and the sugar-acid ratio within the fruit (*Yamaki, 2010*). Consequently, it is crucial to elucidate the molecular mechanisms governing the accumulation of soluble sugars and the metabolism of organic acids in tomato fruits. Sucrose metabolism serves as one of the principal sources of sugar-acid substances in tomatoes. Among the enzymes involved in this metabolic pathway, SUS (sucrose synthase) is a vital one, playing a significant role in sugar accumulation in fruits (*Wang et al., 2018a*). Glycolysis and gluconeogenesis are key metabolic processes through which plants derive energy. In our study, it was found that the soluble sugar and soluble solid contents of the No. 19 yellow tomato were higher than those of the No. 20 red tomato at both the color-break and maturity stages. Furthermore, for both tomato strains, there was a gradual increase in the contents of soluble sugars and soluble solids as the fruits ripened. A similar pattern of sugar accumulation has been observed in studies on cucumbers and melons (*Schemberger et al., 2020*; *Zhang et al., 2016*). In this study, for the No. 19 yellow tomato, the enzymes PFK (phosphofructokinase), FBA (fructose-bisphosphate aldolase), and PK (pyruvate kinase) involved in glycolysis showed a more gradual increase during both the color-break and maturity periods. Additionally, the two FBP (fructose-1,6-bisphosphatase) genes detected in gluconeogenesis followed the same trend. The up-regulation of gene expression related to sugar metabolism in the No. 19 tomato may explain its greater sweetness compared to the No. 20 tomato.

The source of acid in tomatoes is largely influenced by the tricarboxylic acid cycle (TCA) within the mitochondria. Moderate concentrations of acid can enhance the flavor of the fruit, while high acid content typically diminishes its quality (*Gao et al., 2018*). The accumulation, metabolism, and transport of acids during tomato fruit development induce changes in pH within the plastid, which subsequently inhibit or activate the activity of relevant enzymes involved in the metabolism of flavor compounds, further affecting the overall flavor profile (*Wang et al., 2018a*). Our study revealed that the total acid content of tomato No. 19 was lower than that of tomato No. 20 during both developmental periods. Furthermore, both tomato varieties exhibited a decreasing trend in total acid as fruit development progressed. Transcriptomic analyses indicated an upward trend in the expression of *CS* and *SuDH*, which regulate acid synthesis, in yellow tomato No. 19 compared to red tomato No. 20. Despite tomato No. 19 having a lower acid content than tomato No. 20, its expression of genes responsible for acid synthesis was higher than that of tomato No. 20. This discrepancy may be attributed to the fact that the flavor of tomatoes is
determined by the ratio of soluble sugars to acids in the pulp (*Cavaiuolo, Cocetta & Ferrante, 2013*). Even though acid-related gene expression is higher in tomato No. 19, if its sugar content is correspondingly elevated and the sugar-to-acid ratio favors sweetness, then tomato No. 19 will still be perceived as sweeter. Additionally, tomato No. 19 might optimize its sugar-acid ratio through other mechanisms (*e.g.*, ion balance, metabolic regulation, *etc.*) to enhance its sweetness. This finding is consistent with the results reported by *Cheng et al. (2022)*. In summary, sugars and acids exhibit varying degrees of accumulation and degradation as the tomato ripens, and these changes significantly influence the final flavor of the fruit.

Color development serves as a symbol of the ripening process in tomato fruits, with the predominant pigments being carotenoids and flavonoids (*Ballester et al., 2009*). Previous studies have indicated that carotenoid biosynthesis is closely linked to fruit coloration (*Gao et al., 2020*; *Ito et al., 2017*). Additionally, the plant growth regulator ethylene regulates carotenoid synthesis and influences fruit color (*Mou et al., 2016*; *Wu et al., 2018*). The lycopene content of tomato No. 20 was significantly higher than that of tomato No. 19 at the maturity stage, suggesting that the reddish color of tomato No. 20 fruits may be attributed to the accumulation of lycopene in our study. Transcriptomic analysis revealed that the expression of genes encoding the enzymes CrtISO, LCYB, VDE, and NCED1, which are involved in carotenoid synthesis, was up-regulated during fruit development in tomato No. 19 compared to tomato No. 20, while an opposite trend was observed for ZEP. The accumulation of LCYB, VDE, and NCED, along with the down-regulation of ZEP in tomato fruit No. 19, may facilitate the accumulation of β-carotene and xanthaldehyde, resulting in the yellow coloration of tomato fruit No. 19. Octahydro lycopene synthase (PSY) is a key enzyme in the carotenoid biosynthesis pathway (*Arya, Mahajan & Jain, 2000*). Previous research has shown that *PSY* is positively regulated by ethylene during fruit ripening, promoting the accumulation of lycopene in fruits (*Alba et al., 2005*). In this study, the expression levels of *ACS* and *ACO* in No. 19 yellow tomatoes were significantly down-regulated at the color-breaking stage compared to No. 20 red tomatoes, with *PSY* also exhibiting down-regulation at both the color-breaking and maturity stages in No. 19 yellow tomatoes. This down-regulation may be attributed to the significant reduction in ethylene content resulting from the down-regulation of *ACS* and *ACO*, which subsequently led to decreased expression levels of *PSY* and suppressed lycopene accumulation, thereby affecting the coloration of No. 19 tomatoes. These findings align with those of Wu, who demonstrated that blocking ethylene action significantly inhibited the accumulation of both lycopene and β-carotene (*Wu et al., 2018*).

Flavonoids are secondary metabolites that are widely distributed in plants (*Wang et al., 2014*). Research has demonstrated that color changes in plant fruits, such as tomatoes and citrus, are connected with flavonoids (*Kurina et al., 2021*; *Zhang et al., 2020*). In our study, the expression of genes related to flavonoid metabolism in yellow tomato No. 19 exhibited a gradual increase during the development of the tomato. Notably, CYP98A and CCoAMT, which are genes involved in the synthesis of coumaroylquinic acid and feruloyl-coenzyme A, respectively, were significantly up-regulated by 11.78 and 7.71-fold at the maturity stage of yellow tomato No. 19 compared to red tomato No. 20. This

suggests that the up-regulation of genes in the flavonoid biosynthesis pathway may contribute to increased flavonoid levels and color changes in fruits, which contrasts with Ballester's findings (*Ballester et al., 2009*).

## CONCLUSIONS

This study detected significant differences in soluble sugars and other substances between the No. 19 yellow tomato and the No. 20 red tomato during the color-break and ripening stages. To explore the mechanisms underlying the changes in taste and color during the development of these two tomato varieties, transcriptome sequencing was conducted at the color-break and ripening stages. The results indicated that various enzymes related to sugar-acid metabolism, carotenoid metabolism, and flavonoid metabolism may play important roles in the changes of taste and color in these two tomato varieties, and could be the key genes controlling their differences. This study provides new insights into the formation and changes in flavor and sensory quality of these two tomato varieties. Further analysis will be conducted to delve deeper into the transcriptome results, verify the content and function of key genes, and elucidate their regulatory mechanisms.

### Funding

This work was supported by the Key Laboratory of Vegetable Biology of Yunnan Province, College of Landscape and Horticulture, Yunnan Agricultural University (202402AN360008), National Natural Science Foundation of China (NSFC) "Molecular Mechanisms of Anther Development Regulated by SlGIS Transcription Factors in Tomato" (3236180187) and Yunnan Provincial Major Science and Technology Special Program "Selection and Breeding of 4 Major Vegetable Varieties of Yunnan Melon" (202402AE090012). The funders had no role in study design, data collection and analysis, decision to publish, or preparation of the manuscript.

### Grant Disclosures

The following grant information was disclosed by the authors:
Key Laboratory of Vegetable Biology of Yunnan Province, College of Landscape and Horticulture, Yunnan Agricultural University: 202402AN360008.
National Natural Science Foundation of China (NSFC): 3236180187.
Yunnan Provincial Major Science and Technology Special Program: 202402AE090012.

### Competing Interests

The authors declare that they have no competing interests.

### Author Contributions

- Chunmei Guo conceived and designed the experiments, performed the experiments, analyzed the data, authored or reviewed drafts of the article, and approved the final draft.
- Xiuyuan Liu performed the experiments, prepared figures and/or tables, authored or reviewed drafts of the article, and approved the final draft.

- Yingfeng Ding analyzed the data, prepared figures and/or tables, authored or reviewed drafts of the article, and approved the final draft.
- Zhaoyilan He analyzed the data, prepared figures and/or tables, and approved the final draft.
- Songmei Shi performed the experiments, analyzed the data, authored or reviewed drafts of the article, and approved the final draft.
- Yumei Ding conceived and designed the experiments, performed the experiments, prepared figures and/or tables, and approved the final draft.
- Hui Shen conceived and designed the experiments, authored or reviewed drafts of the article, and approved the final draft.
- Zhengan Yang conceived and designed the experiments, authored or reviewed drafts of the article, and approved the final draft.

## Data Availability

The data are available at NCBI GEO: GSE285925.

## Supplemental Information

Supplemental information for this article can be found online at http://dx.doi.org/10.7717/peerj.20113#supplemental-information.

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
