# Peer review of "Transcriptome reveals differential expression of flavor and color in closely related strains of tomato (Solanum lycopersicum)"

_PeerJ, doi:10.7717/peerj.20113_

## Round 0.1 · original submission · Major Revisions

Reviewer 1 ·

Basic reporting

This manuscript examines “Transcriptome reveals differential expression of flavor and color in closely related species of tomato”. This analysis aimed to identify key regulatory genes and biosynthetic pathways related to flavor and color development in tomato fruits. Here, we will put forward some suggestions in the hope of helping authors improve the quality and rigor of their papers:
1. It is recommended that the innovation of the research should be emphasized in the abstract, e.g. whether the comparison of yellow and red tomato lines is the first to be reported, or whether new regulatory genes or pathways have been identified.
2. It is recommended that authors state the period in which the samples were sent, or explain what BR19, BR20; MF19, MF20 stand for before describing the transcriptome results in the abstract.
3. “……as well as genes linked to ethylene synthesis, such as 1-aminocyclopropane-1-carboxylate synthase (ACS) and 1-aminocyclopropane-1carboxylic acid oxidase (ACO), may play a role in the color changes observed in tomatoes”. How are genes related to ethylene synthesis associated with tomato color change, which is not directly reflected in the text, please explain or verify.
4. It is suggested that the authors adjust the format of the paper. Now, there are significant problems with the format of the full text.
5. The Latin name of tomato in the article should appear where the author first mentions tomato.
6. The author identified several key genes in the sugar-acid metabolism pathway in the article. It is suggested that the author should refer to previous literature and describe this part in the preface of the article, as well as why the author chose to identify these enzymes.
7. The methods for determining the total acid, vitamin C and soluble sugar content in the article are described too simply. It is suggested that the author describe the reagents used and the sample treatment methods in more detail in the materials and methods section.
8. The "P" in the captions should be in italics. It is suggested that the author correct this format error throughout the text.
9. In the result section of the article, when the author describes the figures, the format used is inconsistent. One moment it is an abbreviation, and the next it is the full name. It is suggested that the author modify the format uniformly according to the requirements of the journal.
10. It is proposed that the relevant figure on reliability analysis of transcriptome data be placed in the Supplementary Figures.
11. Plots of GO and KEGG analyses of differential genes from the same period as well as from different periods were put together for easy comparison. The other figures that follow have this problem, and it is recommended that the figures be rearranged and combined appropriately for the reader to see them.
12. Figs 4,5,6, please change the color of the gene expression up/down-regulation icons to avoid red and green together as much as possible.
13. Fig 8a (1)(2), same legend. Please confirm it.
14. There are many grammatical issues in the results section of the article. It is suggested that the author unify the tenses and check for grammatical errors.
15. Figures resolutions are too low, please provide a higher resolution figure.

Experimental design

Please see "Basic reporting".

Validity of the findings

Please see "Basic reporting".

Additional comments

No.

Reviewer 2 ·

Basic reporting

Flavor and color are important quality traits of tomato and are the main indicators for consumers to evaluate the quality of tomato through their senses. In this paper, two cultivars with significant differences only in flavor and color were selected as materials. The related physiological indicators were determined, and the screening and identification of DEGs were carried out through transcriptomic analysis. The research content is scientific and rigorous, and the research results can provide a basis for the improvement of quality.

I suggest that this manuscript can be publication after revision.

Comments:

1.Some of the figures in the manuscript are combined together (such as Fig.1a, 1b), but in the review version, all the figures are separate. Please provide the combined version;

2. The captions of the cultivar names in Figure 1b are not in English;

3. Some descriptions of the methods in the Results, such as line 232-236 “We compared…… q < 0.05”, have already been described in the Materials and Methods. It is suggested that the explanations of the methods be reduced in the Results and supplemented in the Materials and Methods;

4. Line 251 “differentially expressed genes (DEGs)”, the abbreviations have been explained in line 237. It is recommended that the abbreviated form still be used subsequently;

5. The names of the GO annotation on the left side of Fig. 3a are incomplete;

6. The title of Fig. 4 is incomplete, the first letters of the titles in Fig. 5 and 6 need to be capitalized;

7. Please add captions for the numbers in the squares in Fig. 4-6;

8. Line 391, 393: “Figure” needs to be replaced with “Fig”;

9. Fig. 9: There is a lack of captions for the corresponding varieties of dark and light colors, and the histogram of qPCR lacks significant difference analysis;

10. Please ensure that the citation format of the references is consistent with the requirements of the journal, such as line 450 “WANG Libin et al., 2017” and line 452 “Su Lulu et al., 2023”;

11. The “Discussion” in lines 441-445 lacks references;

12. The “Conclusions” should concisely summarize the results and new findings of this study, and should not contain any irrelevant content;

13. References need to be checked for complete information, such as line 563 “BALDWIN, E. A., GOODNER, K. & PLOTTO, A. 2008. Interaction of Volatiles, Sugars, and Acids on Perception of Tomato Aroma and Flavor Descriptors. Journal of Food Science”

Experimental design

no comment

Validity of the findings

no comment

Additional comments

no comment

Reviewer 3 ·

Basic reporting

This manuscript is well structured. It is well written and illustrated clearly. The experimental design is straightforward, and the authors assessed two cultivars of tomato fruit by transcriptional analysis. The main results focus on reporting the RNASeq data in detail. However, the results are mostly basic reporting, statistic without deeply devolving into the biological questions.

Experimental design

It's very clear and simple. The background information of the two cultivars should be described. In addition, if there are mapping data supporting the genetic variation between these two cultivars?

Validity of the findings

It meets the standard.

Additional comments

The quality of most figures could be improved. For example, the text is very faint and the width-length ratio of some figures are improper. Some figures are on a same topic, so they should be grouped together.

---

## Round 0.2 · Minor Revisions

**Language Note:** When you prepare your next revision, please either (i) have a colleague who is proficient in English and familiar with the subject matter review your manuscript, or (ii) contact a professional editing service to review your manuscript. PeerJ can provide language editing services - you can contact us at [email protected] for pricing (be sure to provide your manuscript number and title). – PeerJ Staff

Reviewer 1 ·

Basic reporting

The authors made some modifications based on the comments. However, there are still some issues that need further clarification.

Experimental design

-

Validity of the findings

-

Additional comments

1. "5. Tomato is mentioned for the first time in the title, which has been added in the title(43). In scientific writing, the title, abstract, and main body of a paper are designed to be relatively independent, and Latin scientific names and abbreviations must be separately addressed in each section.

2. "12. The red and green folds represent the up-regulated and down-regulated gene folds are more obvious." Red-green color blindness, a common X-linked recessive genetic disorder, affects approximately 4-8% of males worldwide. To ensure inclusivity and minimize inconvenience for individuals with this condition, it is advisable to avoid relying exclusively on red and green colors in visual materials. Instead, consider using alternative color combinations or incorporating additional visual cues (e.g., patterns, labels, or textures) to enhance accessibility.

3."13. The figures of 8a(1) and 8a(2) represent the different transcription...": The mention of Figure 8 here seems erroneous, given that the paper contains only Figures 1 to 6.

Reviewer 2 ·

Basic reporting

I would like to thank the authors for improving the manuscript. In my opinion, the revised manuscript is ready for publication.

Experimental design

-

Validity of the findings

-

---

## Round 0.3 · Minor Revisions

The reviewer has recommended acceptance of the revised manuscript. However, I noticed a problem with wording the title and abstract, both of which refer two "closely related species of tomato". However, No. 19 and No. 20 are not two difference species of tomato (closely related species). Instead, they are varieties or cultivars or strains of one species. Elsewhere in the manuscript, you refer to them as strains. To maintain consistency, the title and abstract would then refer to "two different strains of tomatoes" For cultivated plants, the terms "cultivars" or "varieties" are used more frequently than "strains", but if you prefer to use the term "strain", that is acceptable in my opinion. I expect that minor but important revision can be easily addressed.

Reviewer 1 ·

Basic reporting

Thank you to the authors for their careful revisions and responses. Thank you!

Experimental design

no comment

Validity of the findings

no comment

---

## Round 0.4 · accepted · Accept

Although the requested changes to the title and abstract can be seen in the Tracked Changes document, those changes were NOT made in the final manuscript document or in the Review PDF. Please ensure that the changes made in the Tracked Changes document (using "strain" not "species") are implemented in your final uploaded document used to produced the PDF.